# No-Press Diplomacy from Scratch

**Anton Bakhtin**   **David Wu**   **Adam Lerer**   **Noam Brown**
Facebook AI Research
{yolo,dwu,alerer,noambrown}@fb.com

## Abstract

Prior AI successes in complex games have largely focused on settings with at most hundreds of actions at each decision point. In contrast, Diplomacy is a game with more than $10^{20}$ possible actions per turn. Previous attempts to address games with large branching factors, such as Diplomacy, StarCraft, and Dota, used human data to bootstrap the policy or used handcrafted reward shaping. In this paper, we describe an algorithm for action exploration and equilibrium approximation in games with combinatorial action spaces. This algorithm simultaneously performs value iteration while learning a policy proposal network. A double oracle step is used to explore additional actions to add to the policy proposals. At each state, the target state value and policy for the model training are computed via an equilibrium search procedure. Using this algorithm, we train an agent, *DORA*, completely from scratch for a popular two-player variant of Diplomacy and show that it achieves superhuman performance. Additionally, we extend our methods to full-scale no-press Diplomacy and for the first time train an agent from scratch with no human data. We present evidence that this agent plays a strategy that is incompatible with human-data bootstrapped agents. This presents the first strong evidence of multiple equilibria in Diplomacy and suggests that self play alone may be insufficient for achieving superhuman performance in Diplomacy.

## 1   Introduction

Classic multi-agent AI research domains such as chess, Go, and poker feature action spaces with at most thousands of actions per state, and are therefore possible to explore exhaustively a few steps deep. In contrast, modern multi-agent AI benchmarks such as StarCraft, Dota, and Diplomacy feature combinatorial action spaces that are incredibly large. In StarCraft and Diplomacy, this challenge has so far been addressed by bootstrapping from human data [32, 24, 3, 11]. In Dota, this challenge was addressed through careful expert-designed reward shaping [4].

In contrast, in this paper we describe an algorithm that trains agents through self play without any human data and can accommodate the large action space of Diplomacy, a longstanding AI benchmark game in which the number of legal actions for a player on a turn is often more than $10^{20}$ [24].

We propose a form of deep RL (reinforcement learning) where at each visited state the agent approximates a Nash equilibrium for the stage game with 1-step lookahead search,

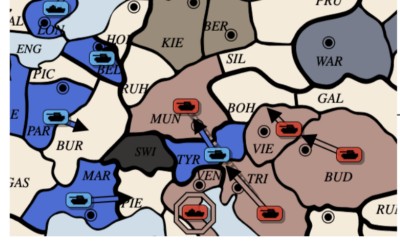

Figure 1: France (blue) would be in a strong position were it not for exactly two actions that Austria (red) can play to dislodge the unit in Tyr. The probability of sampling one of them randomly is roughly $10^{-6}$.

plays the equilibrium policy, and uses the equilibrium value as the training target. Since the full action space is too large to consider, this equilibrium is computed over a smaller set of candidate

35th Conference on Neural Information Processing Systems (NeurIPS 2021).

actions. These actions are sampled from an action proposal network that is trained in unison with the value model. We show that our deep Nash value iteration algorithm, when initialized with a human-bootstrapped model, defeats multiple prior human-derived agents in no-press Diplomacy by a wide margin.

Next, we investigate training an agent completely from scratch with no human data in Diplomacy. In order to improve action exploration when training from scratch, the proposal actions are augmented with exploratory actions chosen through a double-oracle-like process, and if these discovered actions end up as part of the equilibrium then they will be learned by the proposal network. Using this technique, which we call *DORA* (Double Oracle Reinforcement learning for Action exploration), we develop a superhuman agent for a popular two-player variant of Diplomacy. Our agent is trained purely through self-play with no human data and no reward shaping.

We also train a *DORA* agent for the 7-player game of no-press Diplomacy, the first no-press Diplomacy agent to be trained entirely from scratch with no human data. When 6 *DORA* agents play with 1 human-data-based agent, all human-data-based agents perform extremely poorly. However, when one *DORA* agent plays with 6 human-data-based agents, the bot underperforms. These results suggest that self-play in 7-player Diplomacy may converge to equilibria that are incompatible with typical human play. This is in contrast to previous large-scale multi-agent benchmarks like multiplayer poker [9] and validates Diplomacy as a valuable benchmark for learning policies that are compatible with human behavior.

## 2 Background and Related Work

### 2.1 Description of Diplomacy

In this section we summarize the rules of Diplomacy. For a more detailed description, see [24]. No-press Diplomacy is a zero-sum board game where 7 players compete for control on a map of Europe. The map contains 75 locations, 34 of which are *supply centers* (SCs). Each player begins with 3-4 units and 3-4 SCs and a player wins by controlling a majority (18) of the SCs. If no player controls a majority and all remaining players agree to a draw, then the victory is divided between the unelimited players. In case of a draw, we use the **Sum-of-Squares (SoS)** scoring system that defines the score of player $i$ as $C_i^2 / \sum_{i'} C_{i'}^2$, where $C_i$ is the SC count for player $i$.

Each turn during movement, all players simultaneously choose an action composed of one order for every unit they control. An order may direct a unit to hold stationary, to move to an adjacent location, to support a nearby unit's hold or move, to move across water via convoy, or to act as the convoy for another unit. Because a player may control up to 17 units with an average of 26 valid orders for each unit, the number of possible actions is typically too large to enumerate. Additionally, in Diplomacy, often supports or other critical orders may be indistinguishable from a no-op except in precise combinations with other orders or opponent actions. This makes the action space challenging to learn with black-box RL. Actions are simultaneous, so players typically must play a mixed strategy to avoid being predictable and exploitable.

In the popular *France vs. Austria (FvA)* variant of Diplomacy, the number of players is reduced to two, but the victory condition and all game rules, including the number of locations and centers, remain the same. FvA serves as an effective way to isolate the challenge of an astronomically large imperfect-information action space from the challenge of Diplomacy's multi-agent nature.

### 2.2 Stochastic Games

*Stochastic games* [26] are a multi-agent generalization of Markov decision processes. At each state, each player chooses an action and the players' joint action determines the transition to the next state. Formally, at each state $s \in S$ agents simultaneously choose a joint action $\boldsymbol{a} = (a_1, \ldots, a_\mathcal{N}) \in A_1 \times \cdots \times A_\mathcal{N}$, where $\mathcal{N}$ is the number of players. Agents then receive a reward according to a vector function $\boldsymbol{r}(s, \boldsymbol{a}) \in \mathbb{R}^\mathcal{N}$ and transition to a new state $s' = f(s, \boldsymbol{a})$. The game starts at initial state $s^0$ and continues until a terminal state is reached, tracing a trajectory $\tau = ((s^0, \boldsymbol{a}^0), ..., (s^{t-1}, \boldsymbol{a}^{t-1}))$ where $f(s^{t-1}, \boldsymbol{a}^{t-1})$ is terminal. Each agent $i$ plays according to a policy $\pi_i : S \to \Delta A_i$, where $\Delta A_i$ is a set of probability distributions over actions in $A_i$. Given a policy profile $\boldsymbol{\pi} = (\pi_1, \ldots, \pi_\mathcal{N})$, the probability of a trajectory is $p(\tau | \boldsymbol{\pi}) = \prod_{(s, \boldsymbol{a}) \in \tau} p(\boldsymbol{a} | \boldsymbol{a} \sim \boldsymbol{\pi}(s))$, and we define the vector of

expected values of the game rooted at state $s$ as $\boldsymbol{V}(\boldsymbol{\pi}, s) = \sum_{\tau \in T(s)} p(\tau | \boldsymbol{\pi}) \sum_{(s_t, \boldsymbol{a_t}) \in \tau} \gamma^t \boldsymbol{r}(s_t, \boldsymbol{a_t})$ where $T(s)$ is the set of terminal trajectories beginning at $s$. We denote by $\boldsymbol{V}(\boldsymbol{\pi}) := \boldsymbol{V}(\boldsymbol{\pi}, s^0)$ the value of a game rooted in the initial state. We denote by $\boldsymbol{\pi}_{-i}$ the policies of all players other than $i$.

A **best response (BR)** for agent $i$ against policy profile $\pi$ is any policy that achieves maximal expected value. That is, $\pi_i^*$ is a BR to $\pi$ if $V_i(\pi_i^*, \pi_{-i}) = \max_{\pi_i'} V_i(\pi_i', \pi_{-i})$. A policy profile is a **Nash equilibrium (NE)** if all players' policies are BRs to each other. Letting $V_i$ be the $i$th component of $\boldsymbol{V}$, $\boldsymbol{\pi}$ is a NE if:

$$\forall i \, \forall \pi_i' : V_i(\boldsymbol{\pi}) \geq V_i(\pi_i', \boldsymbol{\pi}_{-i})$$

An $\epsilon$**-NE** is a policy profile in which an agent can only improve its expected reward by at most $\epsilon$ by switching to a different policy. If a policy profile is an $\epsilon$-NE, we say its **exploitability** is $\epsilon$.

We may also talk about Nash equilibria for one stage of the game rather than the whole game. For any state $s$, supposing we have values $Q(s, \boldsymbol{a})$ that represent the expected values of playing joint action $\boldsymbol{a}$ in state $s$ assuming some policy profile is played thereafter, we may define a new 1-step game where instead the game ends immediately after this action and $Q(s, \boldsymbol{a})$ is itself the final reward. For such stage games, notationally we denote NE using the symbol $\sigma$ rather than $\pi$.

## 2.3 Nash Q-Learning

Value iteration methods such as Nash-Q have been developed that provably converge to Nash equilibria in some classes of stochastic game, including 2p0s (two-player zero-sum) stochastic games [19, 14]. Nash-Q learns joint Q values $\boldsymbol{Q}(s, \boldsymbol{a})$ that aim to converge to the state-action value of $(s, \boldsymbol{a})$ assuming that some NE $\boldsymbol{\pi}$ is played thereafter. This is done by performing 1-step updates on a current estimated function $\boldsymbol{Q}$ as in standard Q-learning, but replacing the max operation with a stage game NE computation.

Formally, suppose that $\boldsymbol{a}$ is played in $s$ leading to $s'$. A NE $\sigma$, $\sigma_i \in \Delta A_i$, is computed for the stage game at $s'$ whose payoffs for each joint action $\boldsymbol{a}'$ are specified by the current values of $\boldsymbol{Q}(s', \boldsymbol{a}')$. The value of this game to each player assuming this NE is then used as the target values of $\boldsymbol{Q}(s, \boldsymbol{a})$. The update rule is thus:

$$\boldsymbol{Q}(s, \boldsymbol{a}) \leftarrow (1 - \alpha)\boldsymbol{Q}(s, \boldsymbol{a}) + \alpha\Big(\boldsymbol{r}(s, \boldsymbol{a}) + \gamma \sum_{\boldsymbol{a}'} \sigma(\boldsymbol{a}')\boldsymbol{Q}(s', \boldsymbol{a}')\Big), \tag{1}$$

where $\alpha$ is the learning rate, $\gamma$ is the discount factor, and $\sigma(\boldsymbol{a}) := \prod_i \sigma_i(a_i)$ is the probability of joint action $\boldsymbol{a}$.

As presented later in Section 3.1, the foundation of our algorithm is an adaptation of Nash Q-Learning, with several modifications to make it suitable for deep RL and adjusted to use a value function rather than a Q function.

## 2.4 Double Oracle

**Double oracle (DO)** is an iterative method for finding a NE over a large set of actions in a matrix game [22]. It leverages the fact that computing a best response is computationally much cheaper than computing a NE. Starting from a small set of candidate actions $A_i^0 \subseteq A_i$ for each player $i$, on each iteration it computes a NE $\sigma^t$ in the matrix game restricted to the candidate actions $\boldsymbol{A}^t$. Then, at each iteration $t$ for each player $i$ it finds the best response action $a_i^{t+1} \in A_i$ to the restricted NE $\sigma^t$ and creates $A_i^{t+1} := A_i^t \bigcup \{a_i^{t+1}\}$. Formally, $a_i^{t+1}$ is an action in the full action set $A_i$ that satisfies

$$a_i^{t+1} = \arg\max_{a' \in A_i} \sum_{\boldsymbol{a} \in \boldsymbol{A}^t} Q_i(s, a_1, \ldots, a_{i-1}, a', a_{i+1}, \ldots, a_{\mathcal{N}}) \prod_{j \neq i} \sigma_j^t(a_j). \tag{2}$$

DO-inspired methods have been used in a variety of other work for handling large or continuous action spaces [15, 7, 18, 1, 21]. Although in the worst case the equilibrium policy may have support over a large fraction of the action space, in practice in many problems a small subset of actions may suffice, and in this case DO-based methods can be computationally efficient due to being able to restrict equilibrium computation to only a tiny fraction of the total number of actions.

As we will present in Section 3.2, we make use of a DO-like procedure both within the RL loop and at inference time, and show it to be an effective way to introduce novel good actions to the policy.

## 2.5 Regret Matching

Both Nash-Q-Learning and DO require the computation of NEs in matrix games, which we approximate using the iterative **regret matching (RM)** algorithm [5, 13], as has been done in past work [11]. RM is only guaranteed to converge to a NE in 2p0s games and other special classes of games [12]. Nevertheless, past work has found that it produces competitive policies that have low exploitability in important non-2p0s games including six-player poker and no-press Diplomacy [9, 11]. We therefore use RM to approximate a NE $\sigma$ in restricted matrix games during DO.

In RM, each player $i$ has a **regret** value for each action $a_i \in A_i$. The regret on iteration $t$ is denoted $\text{Regret}_i^t(a_i)$. Initially, all regrets are zero. On each iteration $t$ of RM, $\sigma_i^t(a_i)$ is set according to

$$\sigma_i^t(a_i) = \begin{cases} \frac{\max\{0, \text{Regret}_i^t(a_i)\}}{\sum_{a_i' \in \mathcal{A}_i} \max\{0, \text{Regret}_i^t(a_i')\}} & \text{if } \sum_{a_i' \in \mathcal{A}_i} \max\{0, \text{Regret}_i^t(a_i')\} > 0 \\ \frac{1}{|\mathcal{A}_i|} & \text{otherwise} \end{cases} \tag{3}$$

Next, each player samples an action $a_i^*$ from $\mathcal{A}_i$ according to $\sigma_i^t$ and all regrets are updated such that

$$\text{Regret}_i^{t+1}(a_i) = \text{Regret}_i^t(a_i) + v_i(a_i, a_{-i}^*) - \sum_{a_i' \in \mathcal{A}_i} \sigma_i^t(a_i') v_i(a_i', a_{-i}^*), \tag{4}$$

where $v_i(\overline{a})$ is the utility of player $i$ in the matrix game. This sampled form of RM guarantees that the *average* policy over all iterations converges to a $\epsilon$-NE in 2p0s games at a rate of at worst $\mathcal{O}(\frac{1}{\epsilon^2})$ iterations with high probability [17]. In order to improve empirical performance, we use linear RM [8], which weighs updates on iteration $t$ by $t$.[1] We also use optimism [30], in which the most recent iteration is counted twice when computing regret.

# 3 Algorithms

We now describe DORA in detail. DORA simultaneously learns a state-value function and an action proposal distribution via neural networks trained by bootstrapping on an approximate Nash equilibrium for the stage game each turn. A DO-based process is used to discover actions.

## 3.1 Deep Nash Value Iteration

The core of the learning procedure is based on Nash Q-Learning as presented in Section 2.3, but simplified to use only a value function, with adaptations for large action spaces and function approximation.

Approximating a Q-function in a game with an action space as large as Diplomacy is challenging. However, in Diplomacy as in many other domains, the reward $r(s, a)$ depends only on the next state $s' = f(s, a)$ and not on how it was reached, and since the transition function is known and can be simulated exactly, we can redefine $r$ to be a function of just the next state $s'$. In that case, we can rewrite Update Rule 1 (from Section 2.3) in terms of a state value function:

$$\boldsymbol{V}(s) \leftarrow (1 - \alpha)\boldsymbol{V}(s) + \alpha(\boldsymbol{r}(s) + \gamma \sum_{\boldsymbol{a}'} \sigma(\boldsymbol{a}')\boldsymbol{V}(f(s, \boldsymbol{a}'))) \tag{5}$$

Since the state space is large, we use a deep neural network $\boldsymbol{V}(s; \theta_v)$ with parameters $\theta_v$ to approximate $\boldsymbol{V}(s)$. Since the action space is large, exactly computing the Nash equilibrium $\sigma$ for the 1-step matrix game at a state is also infeasible. Therefore, we train a policy proposal network $\pi(s; \theta_\pi)$ with parameters $\theta_\pi$ that approximates the distribution of actions under the equilibrium policy at state $s$. We generate a candidate action set at state $s$ by sampling $N_b$ actions from $\pi(s; \theta_\pi)$ for each player and selecting the $N_c \ll N_b$ actions with highest likelihood. We then approximate a NE $\sigma$ via RM (described in Section 2.5) in the restricted matrix game in which each player is limited to their $N_c$ actions assuming successor states' values are given by our current value network $\boldsymbol{V}(s; \theta_v)$.

We explore states via self-play to generate data, with both players following the computed NE policy with an $\epsilon$ probability of instead exploring a random action from among the $N_c$ proposals. Also, in

---

[1] In practice, rather than weigh iteration $t$'s updates by $t$ we instead discount prior iterations by $\frac{t}{t+1}$ in order to reduce numerical instability. The two options are mathematically equivalent.

place of $\theta_v$ and $\theta_\pi$, self-play uses slightly-lagged versions of the networks $\hat{\theta}_v$ and $\hat{\theta}_\pi$ to compute $\sigma$ that are only updated periodically from the latest $\theta_v$ and $\theta_\pi$ in training, a normal practice that improves training stability [23]. We then regress the value network towards the computed stage game value under the NE $\sigma$ using MSE loss (whose gradient step is simply the above update rule), and regress the policy proposal network towards $\sigma$ using cross entropy loss:

$$\text{ValueLoss}(\theta_v) = \frac{1}{2}\left(\boldsymbol{V}(s;\theta_v) - \boldsymbol{r}(s) - \gamma\sum_{\boldsymbol{a'}}\sigma(\boldsymbol{a'})\boldsymbol{V}\left(f(s,\boldsymbol{a'});\hat{\theta}_v\right)\right)^2$$
$$\text{PolicyLoss}(\theta_\pi) = -\sum_{i}\sum_{a_i \in A_i}\sigma_i(a)\log\pi_i(s,a_i;\theta_\pi),$$

(6)

We show in Appendix B that in 2p0s games, the exact tabular form of the above algorithm with mild assumptions and without the various approximations for deep RL provably converges to a NE. Outside of 2p0s games and other special classes, both Nash Q-learning and RM lack theoretical guarantees; however, prior work on Diplomacy and 6-player poker have shown that similar approaches still often perform well in practice [11, 3, 9], and we also observe that DORA performs well in 7-player no-press Diplomacy.

At a high level our algorithm is similar to previous algorithms such as AlphaZero [28] and ReBeL [6] that update both a policy and value network based on self-play, leveraging a simulator of the environment to perform search. The biggest differences between our algorithm and AlphaZero are that 1-ply RM acts as the search algorithm instead of Monte Carlo tree search (just as in [11]), the value update is uses a 1-step bootstrap instead of the end-of-game-value (which allows the trajectory generation to be off-policy if desired), and the policy network acts as an action proposer for the search but does not regularize the search thereafter. Our algorithm is also similar to a recent method to handle large action spaces for Q-learning [31] and reduces to it in single-agent settings.

## 3.2 Double-Oracle-Based Action Discovery

The algorithm in Section 3.1 relies on the policy network to generate good candidate actions. However, good actions initially assigned too low of a probability by the network may never be sampled and reinforced. To address this, we use a mechanism based on double oracle, described in Section 2.4, to introduce novel actions. However, vanilla DO requires computing the expected value of all actions against the restricted matrix game NE. Since our action space is too large to enumerate, we generate only a pool of $N_p \gg N_c$ actions to consider.

A natural idea to generate this pool would be to sample actions uniformly at random from all actions. However, our results in Section 5.1 show this approach performs poorly. Intuitively, in a combinatorial action space as large as in Diplomacy often only a minuscule fraction of actions are reasonable, so random sampling is exceedingly unlikely to find useful actions (see Fig 1 for an example).

We instead generate the pool of $N_p$ actions for a player by generating local modifications of the candidate actions already in the restricted matrix game equilibrium. We first sample uniformly among the player's $N_d$ ($d \le c$) most probable actions in that equilibrium. Next, for each sampled action, we randomly sample a single location on the map and consider all units of the player adjacent to the location. For those units, we randomly choose legal valid new orders. We add the order in this pool that best-responds to the equilibrium as a new candidate action, and DO proceeds as normal.

As shown in Table 3, we found this method of action sampling to outperform simply picking legal and valid combinations of orders uniformly at random. Intuitively, this is likely because locality in Diplomacy, as in a wide variety of other problem domains and algorithms, is a powerful inductive bias and heuristic. Perturbations of good actions are vastly more likely than random actions to also be good, and spatially-nearby actions are more likely to need to covary than spatially-distant ones.

To further improve the scalability of our algorithm, we apply several additional minor approximations, including computing some of the above expected values using sampling and capping the number of DO iterations we perform. For details, see Appendix D.

# 4 Implementation and Methods

We describe our implementation of the above algorithms and training methods as applied to training in Diplomacy. For details on the exact parameters, see Appendix D. The code and the models are available online [2].

## 4.1 Data Generation and Training

We implement deep Nash value iteration following the same considerations as deep Q-learning. Multiple data generation workers run self-play games in parallel. The policy and value targets on each turn along with the game state they correspond to are collected in a shared experience replay buffer and sampled by a set of training workers to train the networks. The policy proposal and value networks used by the data generation workers are only periodically updated from those in training.

During self play, each agent plays according to its part of the computed approximate-NE policy $\sigma$ with probability $1 - \epsilon$, and with probability $\epsilon$ plays a uniformly random action from the $N_c$ actions obtained from the policy proposal network. We also use larger values of $\epsilon$ on the first two turns to quickly introduce diversity from the opening state, similar to the use of larger early exploration settings in other work, for example AlphaZero [29, 28]. Detailed parameters are in Appendix D.

---

**Algorithm 1** Approximated Double Oracle

1: **function** FINDEQUILIBRIUMWITHDO($s$, $N_c$, $N_p$, $N_{iters}$, $\epsilon$)
2:     $\boldsymbol{A} \leftarrow$ SAMPLEACTIONSFROMPROPOSALNET(s, $N_c$)
3:     $\boldsymbol{\sigma} \leftarrow$ COMPUTEEQUILIBRIUM(s, $\boldsymbol{A}$)
4:     **for** $t = (1..N_{iters})$ **do**
5:         modified $\leftarrow 0$
6:         **for** $i = (1..N_{players})$ **do**
7:             $\hat{A}_i \leftarrow$ GENERATEACTIONS($s$, $A_i$, $N_p$)            ▷ Uniform sampling or local modification
8:             $v_i \leftarrow \mathbb{E}_{\boldsymbol{a} \sim \boldsymbol{\sigma}} [V_i(f(s, \boldsymbol{a}); \theta_v)]$
9:             $a_i^*, v_i^* \leftarrow \arg \max_{a_i \in \hat{A}_i} \mathbb{E}_{\boldsymbol{a}_{-i} \sim \boldsymbol{\sigma}_{-i}} [V_i(f(s, \boldsymbol{a}); \theta_v)]$
10:            **if** $v_i^* - v_i > \epsilon$ **then**
11:                $A_i \leftarrow A_i \bigcup \{a_i^*\}$
12:                $\boldsymbol{\sigma} \leftarrow$ COMPUTEEQUILIBRIUM(s, $\boldsymbol{A}$)
13:                modified $\leftarrow 1$
14:        **if** !modified **then**
15:            Break
        **return** $\sigma$

16: **function** GENERATEACTIONSLOCALMODIFICATION($s$, $A_{base}$, $N_p$)
17:     $A \leftarrow \{\}$
18:     **while** $|A| < N_p$ **do**
19:         $a \leftarrow$ SAMPLEACTION($A_{base}$)
20:         $loc \leftarrow$ SAMPLEMAPLOCATION()
21:         $clique \leftarrow$ GETNEIGHBOURS($loc$)
22:         $a' \leftarrow$ RANDOMCOORDINATEDMODIFICATION($a$, $clique$)
23:         $A \leftarrow A \bigcup \{a'\}$
        **return** $A$

---

## 4.2 Model Architecture

Our model closely follows the encoder-decoder architecture of [11]. We use the policy head to model the policy proposal distribution and the value head to predict the expected values of the players.

We found several adjustments to be effective for our use in deep Nash-based RL. Firstly, we removed all dropout layers, as we found this to improve training stability. Secondly, we replaced the graph convolution-based encoder with a simpler transformer encoder, improving model quality and removing the need to hardwire the adjacency matrix of the Diplomacy map into the architecture. Lastly, we also split the network into separate policy and value networks. Since our regret minimization procedure is essentially only a 1-ply search, the states on which we need to evaluate policy

---

[2] https://github.com/facebookresearch/diplomacy_searchbot

(the current state) and value (the states after one turn) are disjoint, so there is no gain to computing both policy and value in one network query, and we found splitting the networks improved the quality and effective capacity of the models. We measure the impact of these choices in Table 2 and describe the architecture in more detail in Appendix A.

### 4.3 Exploitability Testing

To obtain a metric of the exploitablity of our final agents, we also train additional exploiter agents that rather than attempting to play an equilibrium, instead hold the exploited agent fixed and attempt to converge to a best response. We compare two approaches.

In the first approach, we reimplement the same actor-critic method with sampled entropy regularization previously used in [11] for 7-player no-press Diplomacy and apply it to 2-player FvA. In this approach, the other agent is treated as a black box, reducing the problem to just a standard Markov Decision Process to which we can apply classical RL techniques. Namely, we use Impala [10], an asynchronous version of actor-critic algorithm. Since search agents may take several seconds to act, [11] first trained a policy network to copy the behavior of the search agent and then applied RL to exploit this proxy agent. The proxy agent acts much faster, which enabled generating data faster. In contrast, we optimized the training loop so that it was possible to train to exploit the search agent directly without first generating a proxy policy network agent.

In the second approach, we begin with the policy and value models of the exploited agent and resume deep Nash value iteration, except with a single simple modification, described in Appendix E, that results in learning a best response rather than an equilibrium.

## 5 Results

Diplomacy features two main challenges for AI research: its large combinatorial action space and its multi-agent (non-2p0s) nature. *DORA* is intended to address the former challenge. To measure its effectiveness on just the action-exploration aspect, we first present results in a 2p0s variant of Diplomacy called France vs. Austria (FvA). We show *DORA* decisively beats top humans in FvA.

Next, we present results for 7-player no-press Diplomacy, which has been a domain for much prior research [24, 3, 11]. We train a *DORA* agent from scratch with no human data and test it against pre-trained models from previous work [24, 11][3]. In the 6v1 setting, (six *DORA* agents playing with one copy of another bot) *DORA* wins overwhelmingly against all other agents. However, in the 1v6 setting (one *DORA* agent playing with six copies of another bot), *DORA* does slightly worse than previous search agents that leveraged human data. Our results suggest that *DORA* converges to strong play within an equilibrium (i.e., a "metagame") that is very different from how humans play. Finally, we present results showing that adding our RL algorithm on top of a human-bootstrapped model clearly beats prior bots in both the 1v6 and 6v1 settings.

### 5.1 Results in 2-player France vs. Austria

| | Austria | France | Total |
|---|---|---|---|
| Average score (+variance reduction) | 89%±5% | 66%± 7% | 78%±4% |
| Raw average score | 100%±0% | 73%±12% | 87%±6% |
| # games | 13 | 13 | 26 |

Table 1: Head-to-head score of *DORA* versus top FvA players on `webdiplomacy.net`. The ± shows one standard error. Variance reduction (Appendix C) was used to compute average scores from raw average scores. Note that for raw Austria score, because DORA managed to win *all* 13 games, anomalously the sample standard deviation is technically 0. The true standard deviation is of course larger but cannot be estimated.

In 2-player FvA, we tested our final *DORA* agent against multiple top players of this variant, inviting them to play a series of games against our agent. All players were allowed to play an arbitrary number of practice games against the agent before playing official counted games. Prior to running the games, we also implemented a simple variance reduction method, described in Appendix C.

---

[3]See Appendix D for the details.

In Table 1 we report both the raw and variance-reduced average scores. DORA defeats top human players by a large and highly significant margin, with an average score of 78% after variance reduction. It also wins significantly more than half of games as France, the side considered much harder to play[4]. These results show DORA achieves a level of play in FvA above that of top humans.

We ran several ablations to loosely estimate the relative importance of the various components contributing to our final agent for France vs Austria.

| Architecture | Final Elo | Params |
|---|---|---|
| GraphConv [11] | $898 \pm 13$ | 46.8M |
| TransformerEnc 5x192 | $1041 \pm\ 7$ | 9.3M |
| TransformerEnc 10x224 | $1061 \pm 12$ | 12.4M |
| TransformerEnc 10x224 (Separate) | $1123 \pm 13$ | 24.8M |

Table 2: Effect in FvA of the architecture on final strength after RL, and number of parameters. GraphConv is the architecture achieving SOTA in supervised learning from [11], and similar to other past work [24, 3]. TransformerEnc $b \times f$ replaces the graph-conv encoder with a transformer encoder with $b$ blocks and $f$ feature channels. "Separate" uses separate value and policy networks instead of a shared network with two heads. Elos are based on 100 games against each of several internal baseline agents, including the agents in Table 4.

**GraphConv vs Transformers**

| DO Method During RL Training | Final Elo |
|---|---|
| No DO | $927 \pm 13$ |
| DO, uniform random sampling | $956 \pm 13$ |
| DO, local modification sampling | $1023 \pm 13$ |

Table 3: Effect of DO methods during training on final FvA agent strength. Local modification sampling improves results greatly, whereas uniform random sampling is relatively ineffective. Elos $\pm$ one standard error were computed based on 100 games against each of several baseline agents, including the agents in Table 4.

**Double Oracle at Training Time**    See Table 2. Although not a primary focus of this paper, we found that at least for use in RL, changing our model architecture to use a transformer encoder rather than a graph-convolution encoder in Section 4.2 resulted in significant gains over prior neural network architectures in Diplomacy [24, 3, 11], while using fewer parameters and still being similarly fast or faster in performance. Enlarging the model and splitting the network led to further gains.

See Table 3. We demonstrate the effectiveness of our DO method in finding improved actions during training. DO's ability to discover new strong actions for the policy proposal network to learn makes a big difference in the final agent. Local modification sampling is effective, whereas random sampling is relatively ineffective.

**Double Oracle at Inference Time**    See Table 4. We also demonstrate the effectiveness of our DO method in finding improved candidate actions at inference time. Uniform random sampling among legal actions to generate the action pool for finding best responses only improves the weakest of our baseline agent at inference time, due to being ineffective at finding better actions within the huge action space, whereas local modification sampling improves all baselines at inference time.

### 5.1.1 Exploitability

We implemented and tested both algorithms from Section 4.3 to estimate a lower bound on the exploitability of our agents. We also test a variant where we adapt deep Nash value iteration for exploitation just as in the second method, but disable policy learning - the value network learns to predict states of higher exploitability, but the policy proposal network is fixed and does not learn.

The results against our best agent are in Table 5. We find that despite being of superhuman strength, the agent is still relatively exploitable. In a 2p0s large-action-space setting, this is not necessarily

---

[4]Between agents of similar strength we often observe France winning only about a third of games vs Austria.

| Model used by both players → DO Method used by measured player ↓ | Model1 | Model2 | Model3 |
|---|---|---|---|
| None | $0.5_{\pm 0}$ | $0.5_{\pm 0}$ | $0.5_{\pm 0}$ |
| DO, uniform random sampling | $0.75_{\pm 0.02}$ | $0.48_{\pm 0.02}$ | $0.47_{\pm 0.02}$ |
| DO, local modification sampling | $0.89_{\pm 0.02}$ | $0.59_{\pm 0.02}$ | $0.56_{\pm 0.02}$ |

Table 4: Effect of using DO at inference time in FvA. Each entry is the average score when both players play the column agent, but one player uses a DO variant at inference time. Model1-Model3 are models of increasing quality developed over the course of our research, with the major differences being that Model1 uses a trained value network with a fixed proposal network, Model2 uses GraphConv encoder, and Model3 uses a small Transformer encoder. Stronger agents require better sampling methods to find candidate actions that help.

| | vs DORA | vs DORA w/ inference-time DO |
|---|---|---|
| Actor-Critic PG [11] | $77.0\%_{\pm 0.6\%}$ | - |
| Deep Nash-Derived Exploiter | $90.5\%_{\pm 0.4\%}$ | $85.8\%_{\pm 0.5\%}$ |
| Deep Nash-Derived Exploiter, values only | $65.7\%_{\pm 0.6\%}$ | $60.5\%_{\pm 0.7\%}$ |

Table 5: Win rates for exploiter bots trained with different techniques against our best FvA agent with and without inference-time DO. The $\pm$ shows one standard error. Actor-Critic PG vs DORA with inference-time DO was not run due to computational cost.

surprising. For example in Go, another game with a moderately large action space, there exist gaps of more than 1000 Elo (nearly 100% winrate) between different agents, all of which are superhuman, even with no explicit training for exploitation [27]. Moreover, in imperfect-information games such as Diplomacy, exploiting a fixed agent is much easier than training an agent with low exploitability, since the exploiting agent does not need to precisely balance the probabilities of its actions.

Nonetheless, we also observe that our adaptation of the deep Nash value iteration method greatly outperforms the simpler actor-critic method from [11], and that both value learning and DO-based policy learning are important for its performance, with value learning improving over the baseline from $50\%$ to $60.5\%$, and policy learning improving further to $85.8\%$ winrate over *DORA*.

We also observe that applying DO at inference time reduces the exploitability, from $90.5\%$ using our strongest exploiter agent to $85.8\%$. Even at inference time, DO continues to find and suggest new effective actions that would otherwise be overlooked by the policy proposal network.

## 5.2 Results in 7-player No-Press Diplomacy

We also applied our methods to 7-player no-press Diplomacy, training a *DORA* agent entirely from scratch with no human data in this more-challenging domain. For comparison, we also trained an agent via deep Nash value iteration beginning from policy and value networks first trained on human games. Our best results were achieved when we forced the agent to use the human-initialized policy for action proposal during training games, never updating it, and also stopping the training early - see additional ablations in Appendix H). Since we do not update the action proposal policy used in training games, we also do not apply DO for this agent, and so we refer to this agent as *HumanDNVI-NPU* ("*d*eep *N*ash *v*alue *i*teration, *n*o *p*roposal *u*pdate"). The results are reported in Table 6. Note that as these are 7-player games, the expected score if the agents perform equally well is $1/7 \sim 14.3\%$.

Unlike in 2p0s settings, in a competitive multi-agent game, particularly one such as Diplomacy where agents must temporarily ally and coordinate to make progress, there may be multiple different equilibria that are mutually incompatible [20]. An agent playing optimally under one equilibrium may perform very poorly in a pool of agents playing a different equilibrium.

For example in Diplomacy, it may be mutually benefit two agents to ally to attack a third, but in such cases players must often coordinate on a choice, for example on which unit attacks and which unit supports, or on which region to attack. Since an order and a support must exactly match to be effective, each choice may be a stable equilibrium on its own. But an agent that plays one choice in a population of agents that always play another in that situation will fail to match and will do poorly.

| 1x ↓ vs 6x → | DipNet [24] | SearchBot [11] | DORA | HumanDNVI-NPU |
|---|---|---|---|---|
| DipNet [24] | - | $0.8\%_{\pm 0.4\%}$ | $0.0\%_{\pm 0.0\%}$ | $0.1\%_{\pm 0.0\%}$ |
| SearchBot [11] | $49.4\%_{\pm 2.6\%}$ | - | $1.1\%_{\pm 0.4\%}$ | $0.5\%_{\pm 0.2\%}$ |
| *DORA* | $22.8\%_{\pm 2.2\%}$ | $11.0\%_{\pm 1.5\%}$ | - | $2.2\%_{\pm 0.4\%}$ |
| *HumanDNVI-NPU* | $45.6\%_{\pm 2.6\%}$ | $36.3\%_{\pm 2.4\%}$ | $3.2\%_{\pm 0.7\%}$ | - |
| DipNet-Transf | $23.4\%_{\pm 2.2\%}$ | $2.1\%_{\pm 0.7\%}$ | $0.0\%_{\pm 0.0\%}$ | $0.3\%_{\pm 0.1\%}$ |
| SearchBot-Transf | $48.1\%_{\pm 2.6\%}$ | $13.9\%_{\pm 1.7\%}$ | $0.5\%_{\pm 0.3\%}$ | $0.9\%_{\pm 0.2\%}$ |

Table 6: SoS scores of various agents playing against 6 copies of another agent. The ± shows one standard error. Note that equal performance is $1/7 \approx 14.3\%$. The non-human DORA performs very well as a 6x opponent, while performing much worse on the 1x side with 6 human-like opponents. The last 2 rows are equivalent to first 2 rows, but are retrained with the same architecture as self-play agents, i.e., TransformerEnc 5x192, showing the effect of architecture on these baselines separately from our deep Nash RL methods.

| 1x ↓ vs 6x → | DORA | DORA (alternate run) |
|---|---|---|
| *DORA* | - | $7.5\%_{\pm 1.0\%}$ |
| *DORA* (alternate run) | $3.2\%_{\pm 0.7\%}$ | - |

Table 7: SoS scores between two runs of DORA, 1 agent versus 6 copies of the other. The ± shows one standard error. Note that equal performance would be $1/7 \approx 14.3\%$.

Our results are consistent with *DORA* converging roughly to an equilibrium that is very different than the one played by humans, but playing well within that equilibrium. *DORA* achieves only about an 11% winrate in a 1-vs-6 setting against one of the strongest prior human-data-based agents, SearchBot [11] (tying would be a score of $1/7 \approx 14.3\%$). However, SearchBot fares even worse with only a 1.1% winrate in a 1-vs-6 setting against *DORA*.

Against a second DORA trained with a different random seed (Table 7), each DORA scores only about 3-8% against the other 1-vs-6. This suggests that different DORA runs may converge to different equilibria from each other.

Our results with *HumanDNVI-NPU* are also consistent with reinforcement learning converging to inconsistent equilibria. *HumanDNVI-NPU* soundly defeats SearchBot, the prior SOTA, winning 36.3% of games 1-vs-6, while SearchBot wins only 0.5% in the reverse matchup. Starting from a human-like policy, deep Nash value iteration massively improves the strength of the agent. Yet *HumanDNVI-NPU* still does extremely poorly vs *DORA*, and vice versa. Our results seem to suggest that in 7-player no-press Diplomacy, deep Nash value iteration converges to an equilibrium in which outside agents cannot compete, even agents trained independently by the same method, but that which equilibrium is found depends on initialization, with some equilibria performing far better versus human-like players than others. See also Appendix H for additional ablations and experiments.

# 6 Conclusions and Future Work

Diplomacy has been a benchmark for AI research for two main reasons: its massive combinatorial action space and its multi-agent nature. We show that *DORA* achieves superhuman performance in 2-player Diplomacy without any human data or reward shaping, despite the combinatorial action space and imperfect information. *DORA* therefore effectively addresses the first challenge of Diplomacy.

Additionally, we show that *DORA* in full-scale 7-player no-press Diplomacy approximates an equilibrium that differs radically from human-data-based agents. When one human-data-based agent plays against a population of *DORA* agents, the one agent loses by a wide margin. Similarly, one *DORA* agent against a population of human-data-based agents does not exceed an average level of performance. These 7-player results provide evidence that self-play from scratch may be insufficient to achieve superhuman performance in Diplomacy, unlike other multiplayer games such as 6-player poker [9]. Furthermore, the results suggest that there may be a wide space of equilibria, with some differing dramatically from human conventions. *DORA* provides the first competitive technique for exploring this wider space of equilibria without being constrained to human priors.

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
