# A   Neural Network Architecture

In this appendix, we describe the neural network architecture used for our agents.

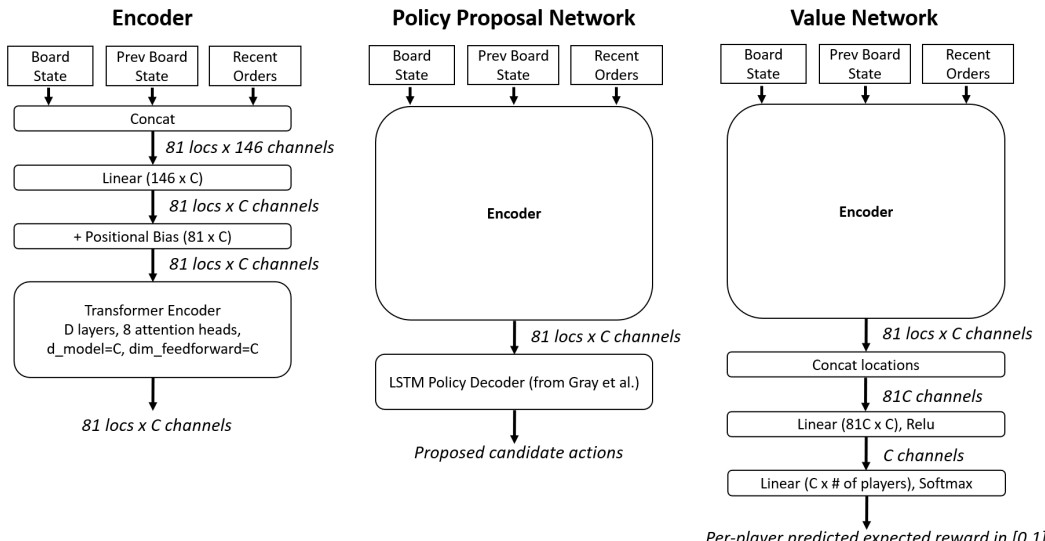

Figure 2: Transformer encoder (left) used in both policy proposal network (center) and value network (right).

Our model architecture is shown in Figure 2. It is essentially identical to the architecture in [11], except that it replaces the specialized graph-convolution-based encoder with a much simpler transformer encoder, removes all dropout layers, and uses separate policy and value networks. Aside from the encoder, the other aspects of the architecture are the same, notably the LSTM policy decoder, which decodes orders through sequential attention over each successive location in the encoder output to produce an action.

The input to our new encoder is also identical to that of [11], consisting of the same representation of the current board state, previous board state, and a recent order embedding. Rather than processing various parts of this input in two parallel trunks before combining them into a shared encoder trunk, we take the simpler approach of concatenating all features together at the start, resulting in 146 feature channels across each of 81 board locations (75 region + 6 coasts). We pass this through a linear layer, add pointwise a learnable per-position per-channel bias, and then pass this to a standard transformer encoder architecture.[6]

In our final agents, we use $(D, C) = (10, 224)$ although in Section 5.1 we also present results for $(D, C) = (5, 192)$ as well as for using a single combined network (i.e. shared encoder) for both policy and value.

# B   Theoretical Correctness of Nash Value Iteration

In any 2p0s stochastic game with deterministic[7] transition function $f$, we show the Deep Nash Value Iteration (DNVI) procedure described in Section 3.1 applied over a candidate set of *all* legal actions converges to a value function $\hat{V}(s)$ that is the minimax value of each reachable state $s$, in the tabular setting. In other words, the use of a value function rather than a Q function does not affect the correctness.

To demonstrate this, we first show an equivalence between our procedure and Nash Q-learning in 2p0s games. At each step of DNVI, we update the value of $V(s)$ using the value computed by RM;

---

[6]For example, `https://pytorch.org/docs/stable/generated/torch.nn.TransformerEncoder.html`

[7]We believe this result holds for games with non-deterministic transitions as well, but that does not allow for a direct reduction from Nash Q learning, so we do not provide a proof here.

since this stage game is 2p0s, RM converges to its unique minimax equilibrium value[13]. For each $(s, \boldsymbol{a})$,

$$\boldsymbol{Q}(s, \boldsymbol{a}) = \boldsymbol{r}(f(s, \boldsymbol{a})) + \boldsymbol{V}(f(s, \boldsymbol{a})),$$

therefore each time $\boldsymbol{V}$ is updated at state $s$ it is equivalent to applying a Nash-Q update at all $(s', \boldsymbol{a})$ for which $f(s', \boldsymbol{a}) = s$.

[14] proves that Nash Q-learning converges to the Q values of a Nash equilibrium for the full game in the tabular setting under the following assumptions:

1. Every one-step game has an adversarial equilibrium and this equilibrium is used in the update rule.

2. All states are visited an infinite number of times.

3. The learning rate $\alpha$ decays such that $\alpha \to 0$ but $\sum \alpha = \infty$.

If these assumptions are satisfied, then $\boldsymbol{V}(s)$ converges to the values of a Nash equilibrium.

Regarding assumption 1, in 2p0s all NEs are automatically adversarial since there is only one opponent (*adversarial* equilibria are NEs that are additionally robust to joint deviations by opponents [14]).

Assumption 2 is satisfied as long as $\forall a, P(a|s) > 0$, which DNVI achieves via $\epsilon$-Nash exploration over the candidate set of all legal actions, which plays a random action with probability $\epsilon > 0$. Note that the modification we use in practice explores only actions that are likely under the current policy proposal net.

Assumption 3 concerns the learning rate schedule, which can be set arbitrarily. In the Deep RL setting we find in practice a fixed learning rate with the Adam [16] optimizer works well.

Thus, DNVI converges to a NE in a 2p0s stochastic game. In a 2p0s game all equilibria have identical (minimax) values. Therefore playing a minimax policy in each stage game based on these values is optimal.

We remind the reader that in Diplomacy the action space is too large to perform exact RM over the entire action space, so in practice we perform DNVI on a small subset of candidate actions proposed by the policy network and rely on Double Oracle to discover actions that should be added to this set.

## C Variance Reduction in Diplomacy

We apply a simple form of variance reduction for our experiments playing with humans to help compensate for the relatively low sample size.

As in many simultaneous-action games, not uncommonly in Diplomacy there are situations that behave like a "matching pennies" subgame, where if player A plays action $a_1$, player B prefers to match it with an action $b_1$, and if A plays action $a_2$, player $B$ prefers to match it with action $b_2$, whereas player $A$'s preferences are the other way around, preferring the outcomes $(a_1, b_2)$ and $(a_2, b_1)$. Rock-paper-scissors-like or other kinds of simple subgames may also occur. In any of these cases, both players should randomize their strategies to reduce exploitability. See Table 8 for an example of a matching-pennies-like subgame.

|       | $b_1$  | $b_2$  |
|-------|--------|--------|
| $a_1$ | -1, 1  | 2,-2   |
| $a_2$ | 1,-1   | -3, 3  |

Table 8: Example payoff matrix for a matching-pennies-like game with nonuniform payoffs. In each cell, the payoff to the row player is listed first.

In such a subgame, once a player commits to playing a particular mixed strategy, then upon seeing the opponent's action, even without knowing the opponent's probability of playing that action, one can identify a significant component of variance due to the luck of randomization in one's *own* mixed strategy that can be subtracted out. For example, in the example in Table 8 above if player A commits to a policy 60% $a_1$ and 40% $a_2$, then supposing player B is revealed to have chosen $b_2$,

player A knows at that point that their expected value is $60\% \cdot 2$ (the payoff for $a_1$) $+40\% \cdot -3$ (the payoff for $a_2$). The deviation from the expected value due to whether player A *actually* samples $a_1$ or $a_2$ on that round is now a matter of pure luck, i.e. just zero-mean variance that can be subtracted out.

In Diplomacy, we can apply this idea as well. When playing a game as agent $i$, on each turn we record the full distribution of our approximated NE $\sigma_i$ from which we sample the action $a_i$ that we submit that turn. Upon observing all other agents' actions $a_{-i}$, and letting $Q$ be the learned approximate Q-value function of our agent, we compute:

$$\delta(a_i) = Q(s, (a_i, a_{-i})) - E_{a'_i \sim \pi_i} Q(s, (a'_i, a_{-i}))$$

$\delta(a_i)$ measures how lucky we are to have chosen the specific $a_i$ that we did on that turn relative to the expected value over all actions $a'_i$ that we could have sampled from policy $\sigma_i$, given the observed actions of the other players. On each turn, we subtract this quantity from the final game win/loss reward of 1 or 0 that we ultimately observe. Since $E_{a_i}\delta(a_i) = 0$, doing this introduces no bias in expectation. So letting $\delta_t$ be the $\delta(a_i)$ value computed for turn $t$, and $R$ the total reward of the game, our final variance-adjusted result is

$$R - \sum_t \delta_t$$

In the event that $Q$ is a good estimate of the true value of the state given the players' policies, then the $\delta_t$ values should be correlated with the final game outcome, so subtracting them should reduce the variance of the outcome, particularly in a 2-player setting such as FvA Diplomacy. In practice, we observe roughly a factor of 2 reduction in variance in informal test matches. As part of our internal practice, we also implemented and tested and committed to the use of this form of variance reduction prior to assembling the final human results.

## D    Implementation details

We report optimizations for Double Oracle we use in practice, details of the pretraining procedure, modifications for 7p, and the hyper parameters used.

### D.1    Double Oracle optimizations

We apply several modifications and approximations to DO to make it run faster in practice:

- We find a best response (line 8 of the algorithm 1) for only one player at a time. When $N_p$ is large, evaluating $N_p$ potential best responses is more expensive than computing a NE among the $N_c$ candidate actions ($N_p \times N_c$ value function calls versus $N_c \times N_c$).[8] Recomputing the NE cheaply ensures each best response takes into account the newest action added by the opponent. Note, that for 7p caching is not as efficient due to large joint action space ($N_c^7$ instead of $N_c^2$), but computing BR still dominates the computations cost.

- When finding a best response, we truncate the opponent's equilibrium policy, which spans up to $N_p$ actions, to only its $k$ highest-probability actions for some small $k$ and renormalize. This allows us to compute a best response with only $N_c * k$ value function calls rather than $N_c * N_p$ calls, while only introducing minimal error.

- We cap the number of iterations of DO we execute.

### D.2    Pretraining

We perform a short phase of pretraining before switching to deep Nash value iteration that improves the speed and stability of training. This phase differs from the main training phase in the following ways:

---

[8]When querying the value network is the dominating cost, the number of iterations needed to compute a NE is largely irrelevant because the value network results can be cached. For this reason, we consider the cost to be roughly $N_c \times N_c$ rather than $T \times N_c \times N_c$.

- Rather than query the policy proposal network and/or apply DO to obtain actions, we select $N_c$ candidate actions for each player uniformly at random among all legal actions.

- Rather than training the value network to predict the 1-step value based on the computed equilibrium over the $N_c$ actions, we train it to directly predict the final game outcome.

By directly training on the final game outcome, this pre-training phase initializes the value network to a good starting point for the main training more quickly than bootstrapped value iteration. This phase also initializes the policy proposal network, which is trained to predict the output of regret matching on the random sampled actions, to a good high-entropy starting point - e.g. to predict a wide range of actions, mildly biased towards actions with high value. If instead a randomly initialized (untrained) policy proposal network were used to select actions, it may select a highly non-uniform initial distribution of candidate actions and require a lot of training for excluded actions to be rediscovered via DO.

### D.3 Extending to 7p

We found a small number of additional implementation details were needed to handle 7-player no-press Diplomacy that were not needed for 2-player FvA.

Firstly, exactly computing the value loss in equation 6 is not feasible because the number of possible joint actions $a'$ scales with the power of the number of players, which for 7 players is too large. So we instead approximate the 1-step value via sampling.

Similarly, during DO, when computing the expected value of potential best responses, given a certain number of actions per opponent, computing the value exactly scales with the power of the number of opponents, which for 6 opponents is too large. So again, we approximate by sampling.

Finally, in line with [11] we found that using Monte Carlo rollouts to compute state values improved the play compared to using a value function alone. Therefore, we use rollouts of depth 2 in all experiments at inference time. We do not use rollouts during training due to the computational cost.

### D.4 Hyper-parameters

### D.4.1 Double Oracle

We use slightly different parameters at training and inference time to speed up the data generation for training. See table 9.

|  | Training | Inference |
| --- | --- | --- |
| Pool size ($N_p$) | 1,000 | 10,000 |
| Max opponent action ($k$) | 8 | 20 |
| Min value difference ($\epsilon$) | 0.04 | 0.01 |
| Max iterations ($N_{iters}$) | 6 | 16 |
| Pool recomputed after each iteration | No | Yes |

Table 9: Hyper-parameter values used for Double Oracle for *DORA*.

### D.4.2 DORA training

Details for training of FvA *DORA* bot are provided in table 10. We use a few additional heuristics to facilate training that are explained below.

The training is bottlenecked by the data generation pipeline, and so we use only a few GPUs for training, but an order of magnitude more for data generation. To make sure the training does not overfit when the generation speed is not enough, we throttle the training when the training to generation speeds ratio is above a threshold. This number could be interpreted as a number of epochs over a fixed buffer.

Our final FvA *DORA* model is trained using 192 Nvidia V100 GPUs on an internal cluster, 4 used for the training and the rest is used for data generation. Both stages take around a week to complete.

We made a number of changes for 7p training compared to FvA for computational efficiency reasons. Each 7p game is 4-5 times longer and each equilibrium computation requires 3 times more operations due to the increased number of players. Moreover, the cost of double oracle also increases many fold as the probability of finding a deviation for at least one player during a loop over players increases. Therefore, for 7p *DORA* we add an additional pre-training stage where we train as normal, but without DO. For speed reasons, we use a smaller transformer model for both the value and the policy proposal nets.

| | FvA | 7p |
|---|---|---|
| Learning rate | $10^{-4}$ | |
| Gradient clipping | 0.5 | |
| Warmup updates | 10k | |
| Batch size | 1024 | |
| Buffer size | 1,280,000 | |
| Max train/generation ratio | 6 | |
| Regret matching iterations | 256 | |
| Number of candidate actions ($N_c$) | 50 | |
| Max candidate actions per unit | - | 6 |
| Number of sampled actions ($N_b$) | 250 | |
| Nash explore ($\varepsilon$) | 0.1 | 0.1 |
| Nash explore, S1901M | 0.8 | 0.3 |
| Nash explore, F1901M | 0.5 | 0.2 |

Table 10: Hyper-parameter values used to train *DORA* agents.

### D.4.3 Evaluation details

We describe parameters used for agent evaluation in table 6.

To compare against DipNet [24] we use the original model checkpoint[9] and we sample from the policy with temperature $0.5$. Similarly, to compare against SearchBot [11] agent we use the released checkpoint[10] and agent configuration[11]. To make the comparison more fair, we used the same search parameters for *DORA* and *HumanDNVI-NPU* as for SearchBot (see table 11).

| | |
|---|---|
| Number candidate actions ($N_c$) | 50 |
| Max candidate actions per unit | 3.5 |
| Number CFR iterations | 256 |
| Policy sampling temperature for rollouts | 0.75 |
| Policy sampling top-p | 0.95 |
| Rollout length, move phases | 2 |

Table 11: Parameters used for all 7p agents with search in table 6 in the main text.

### D.4.4 Distributed training and data generation

We use PyTorch [25] for distributed data parallel training and a custom framework for distributed data generation. We run several Python processes in parallel across multiple machines:

- **Training processes.** All training processes run on one machine. Each is assigned a separate GPU and has a separate replay buffer from which it computes gradients, which are then broadcast and synchronized and across processes. One training process is also responsible for publishing new checkpoints for data generation workers to use and for collecting training statistics.

---

[9] DipNet SL from `https://github.com/diplomacy/research`. MIT License

[10] blueprint from `https://github.com/facebookresearch/diplomacy_searchbot/releases/tag/1.0`. MIT License

[11] `https://github.com/facebookresearch/diplomacy_searchbot/blob/master/conf/common/agents/searchbot_02_fastbot.prototxt`

- **Data collection and rollout processes.** Each machine other than the training machine has a data collection process and many rollout processes. The data collection process gathers rollouts from rollout processes on the same machine via shared memory and sends them to the replay buffers via RPC. Rollout processes run the data generation loop. Each iteration of the loop consists of reading a new model checkpoint if available, stepping the game till the end, and sending results to the collection process. We run up to 8 rollout processes per GPU. Each process has its own copy of both the model and the environment.

- **Evaluation processes** run one-vs-six games of the current model checkpoint versus some fixed agent and collect running winrates for monitoring the run.

- **The metric collection process** receives metrics from other processes via RPC, aggregates and saves them.

Our design was optimized to work on 8-GPU machines with 10 CPUs per GPU, such as Nvidia DGX-1 machines. Our main DORA run used 192 GPUs total, across 24 machines. In such a setting our setup achieves over 90% average utilization for both GPU and CPU.

## E  Adapting Deep Nash Value Iteration to Learn Best Responses

As mentioned in Section 4.3, our best exploiter agents resulted from adapting our main deep Nash value iteration to learn a best response instead of an equilibrium.

To do so, we begin with the policy and value models of the exploited agent and resume deep Nash value iteration, except with a simple modification. On each turn of the game we first invoke the exploited agent to precompute its policy. Then the exploiting agent, when performing Nash learning, only performs RM for itself while the opponent samples only the precomputed policy. All other aspects of the architecture are identical.

Whereas normal RM approximates an NE, one-sided RM instead approximates a best response. With this one change, deep Nash value iteration, rather than training the networks to learn equilibrium policy and values, trains them to learn best-response exploitation and expected state-values assuming future best-response exploitation.

This second approach resembles the Sampled Best Response (SBR) exploiter from [3], in that both methods use a policy model to sample candidate actions for the exploiter, and then approximate a 1-ply best response by directly maximizing against the exploited-agent's known policy with respect to a value model. However, unlike SBR, which uses policy and value models obtained via other methods such as supervised learning that may not be optimized for exploitation, our approach directly trains the policy to sample better exploitative candidate actions, and trains the value to also prefer states where exploitability on future turns will be high.

At test time, our exploiter agent takes the average of 3 samples of the exploited agent's reported average policy instead of only one to get a better estimate, since due to sampling noise or multiple equilibria, regret matching will sometimes return different final average policies. The exploited agent in test itself acts according to a 4th entirely independent sample, to ensure the exploiter can only optimize against the average and not the exact seed.

## F  Human data

To conduct human experiments we informally reached out to top FvA Diplomacy players with a proposal to help with our research by playing against our agent.

The message contained the following instructions:

```
Hi [Person],

I'm AUTHORNAME, one of the developers of BOTNAME. We've put a new version
of the bot online recently that we think might be superhuman, and we're
hoping to do some testing against top humans to measure whether that's the
case.
```

```
Would you be interested in playing  10 games against the bot?  You'd be
able to play at your own pace.  You're also welcome to play against the bot
as many times as you'd like as practice before starting the "real" matches.

For the real matches, you could call them something like "[Person] vs
BOTNAME 1", or something along those lines.  You can launch them just like
a normal game.  We just ask that you not cancel any games and only put in a
draw if it's clearly a stalemate.

These results (aggregated with other top players) would go into an academic
paper.  We could either use your real name or your username, or just report
your FvA GhostRating, whichever you prefer.
```

Among the players that we contacted, 5 played matches against our bot. Those players were ranked 1st, 8th, 16th, 22nd, and 29th in FvA among all players on `webdiplomacy.net` according to the Ghost-Ratings [2] ranking system. None of the individual players had a positive win rate against the bot, though no single player played enough games against the bot to measure individual performance with statistical significance.

We did not collect any information regarding the games besides player's actions in the games played. Thus, the data does not contain any personally identifiable information or offensive content.

## G   Action Exploration Example

We go into more detail regarding the example provided in Figure 1 that motivated our double oracle approach. This is an example of a situation from a real game played by our agent (France, blue) vs a human (Austria, red). The agent has an army in Tyr next to the SCs of the opponent. None of the actions that the agent considers for red could dislodge the army *and* block its retreat further east. Therefore, the estimated state value for blue in this position is high and one of the possible expected outcome is shown in Figure 3.

However, human players were quick to realize that there are two actions that can dislodge the blue army and force it to retreat to Pie (Figure 4, left). Moreover, as the agent blocked its own retreat by moving Mar to Pie, the agent had to disband its army in Tyr (Figure 4, right). Once the optimal action for red is added to the set of candidate actions, the probability of blue moving Mar to Pie and the expected score for blue in this state go down.

There are more than 4 million valid actions[12] or around 400,000 valid coordinated actions. While it is still computationally feasible to run exact DO at inference time for this number of actions, adding even one more unit makes DO search infeasible. Moreover, running DO at inference time only is not sufficient, as during optimal play an agent should know that this state has low value and so the actions leading to this state should be avoided. Since the number of actions one can realistically evaluate during training is around 1,000, we have less than a 1% chance of finding a BR action at training time. Our approximate local modification approach allows us to dramatically reduce the search size and find good approximate BRs with relatively few actions.

Our final *DORA* agent finds red's action from Figure 4 without using inference time DO, as the action is learned by the policy proposal network.

---

[12]Product of possible orders per location: MUN (17), SEV (6), VEN (8), VIE (18), TRI (18), BUD (16). Unit in SEV is not shown on the map.

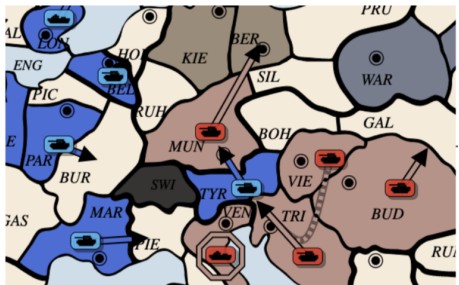 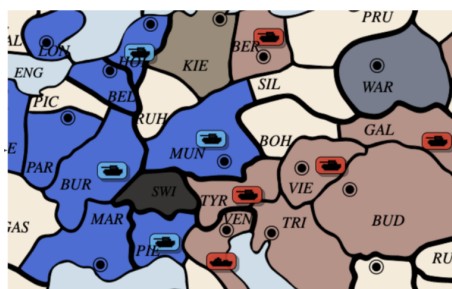

Figure 3: The most probable actions for both players as predicted by the agent. The left figure shows the actions and the right figure shows the state after the actions are executed. France's army escapes to MUN.

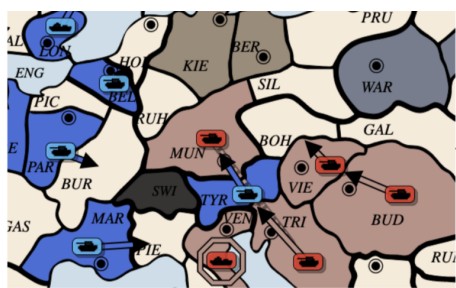 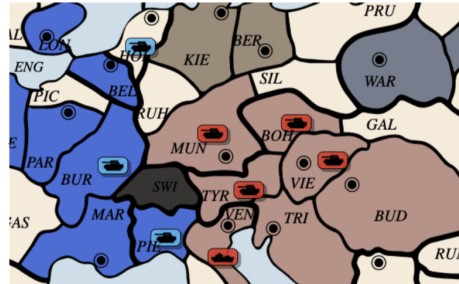

Figure 4: The most probable action for France (blue) as predicted by the agent vs the best action for Austria (red) that the policy proposal network failed to find on its own. The left figure shows the actions and the right figure shows the state after the actions are executed. France's army is crushed and disbanded.

## H Additional Results for 7-player No-Press Diplomacy

In this section we include more preliminary results exploring how initialization and training affect convergence to different equilibria for *HumanDNVI-NPU*.

Namely, we show in Table 12 how performance versus human-like agents changes as self-play training progresses as well as the effect of freezing the action proposal network ("no policy update", "NPU"). As a reminder, *HumanDNVI-NPU* uses a training procedure identical to *DORA*, except a fixed blueprint from supervised learning on human games is used to propose actions throughout the training and hence no DO is used. The policy proposal net is still trained, but is only used at test time. *HumanDNVI* is one step closer to *DORA*: it does update the policy proposal network during training on rollout workers, but does not use DO. In general, both early stopping and NPU help the agent converge towards a strategy more effective against human-like models.

Additionally, we evaluated *HumanDNVI-NPU*-BP-Policy, an agent that uses the value function from *HumanDNVI-NPU*, but the human blueprint policy proposal net during both training and test time. At all times this model performs worse than *HumanDNVI-NPU*. This suggests that a trained policy does increase the strength of the agent, even though using that policy within the RL loop makes the training too-easily diverge away from human-compatible strategies or equilibria. More research in the future may find better techniques to get the best of both worlds.

Finally, Table 13 provides additional data on the impact of DO for 7-player Diplomacy. We trained an agent from scratch for 300k updates without DO and then trained for 60k updates more with DO. This finetuning doubles the score versus SearchBot and also greatly increases it versus Dipnet.

| 1x Agent | # training updates | vs 6x DipNet [24] | vs 6x SearchBot [11] |
|---|---|---|---|
| DipNet [24] | - | - | $0.8\%_{\pm0.4\%}$ |
| Transf | - | $23.4\%_{\pm2.2\%}$ | $2.1\%_{\pm0.7\%}$ |
| SearchBot [11] | - | $49.4\%_{\pm2.6\%}$ | - |
| Transf+Search | - | $48.1\%_{\pm2.6\%}$ | $13.9\%_{\pm1.7\%}$ |
| *DORA* | 600k | $22.8\%_{\pm2.2\%}$ | $11.0\%_{\pm1.5\%}$ |
| *HumanDNVI* | 100k | $30.6\%_{\pm2.4\%}$ | $20.5\%_{\pm2.0\%}$ |
| *HumanDNVI* | 300k | $25.3\%_{\pm2.3\%}$ | $18.6\%_{\pm2.0\%}$ |
| *HumanDNVI-NPU* | 50k | $45.6\%_{\pm2.6\%}$ | $36.3\%_{\pm2.4\%}$ |
| *HumanDNVI-NPU* | 100k | $41.4\%_{\pm2.5\%}$ | $34.3\%_{\pm2.3\%}$ |
| *HumanDNVI-NPU* | 300k | $35.9\%_{\pm2.6\%}$ | $28.4\%_{\pm2.3\%}$ |
| *HumanDNVI-NPU*-BP-Policy | 50k | - | $25.7\%_{\pm2.1\%}$ |
| *HumanDNVI-NPU*-BP-Policy | 100k | - | $25.0\%_{\pm2.1\%}$ |
| *HumanDNVI-NPU*-BP-Policy | 300k | - | $22.9\%_{\pm2.0\%}$ |

Table 12: SoS scores of various agents playing against 6 copies of another agent. The $\pm$ shows one standard error. Note that equal performance is $1/7 \approx 14.3\%$. All our agents are based on TransformerEnc 5x192.

| 1x Agent | # training updates | vs 6x DipNet [24] | vs 6x SearchBot [11] |
|---|---|---|---|
| *ScratchDNVI* | 300k | $13.6\%_{\pm1.8\%}$ | $3.5\%_{\pm0.9\%}$ |
| +finetune with DO | +60k | $21.9\%_{\pm2.2\%}$ | $8.0\%_{\pm1.4\%}$ |

Table 13: Effect of using DO for training from scratch. Even training for a small fraction of time with DO increases the strength of an agent by a significant margin.