# OpenReview forum: "No-Press Diplomacy from Scratch"
_NeurIPS.cc/2021/Conference — NeurIPS 2021 Poster_

### Official Review · Reviewer_GFW6 · 2021-07-08

**Rating:** 6
**Confidence:** 5

**Summary:**

The paper proposes an approach called DORA for building agents for the Diplomacy board game, which is a long standing AI challenge. The key achievement in the paper is constructing agents in a way that does not rely on imitating human gameplay (as opposed to some recent works which at least in part leverages historical game data). Diplomacy is hard because it has a large action space and as it is a game with many players (7 in the full game, but only two int eh France/Austria variant). The results are very good for the two player (France/Austria) variant against people. The results are more mixed in the 7 player game - when there are many copies of DORA, it does very well, but less well when there are many copies of another bot and one DORA player.

**Limitations And Societal Impact:**

The paper adequently adresses this (really this is a board game, I do not see how a good AI for playing a board game would have negative societal impact).

**Main Review:**

Diplomacy is a great AI research domain, and there has been several papers on this challenge. The authors do discuss what makes the game hard (in particular a large action space and having many players). However, Diplomacy is just a board game, so I think the authors should extend the discussion of why this domain is so important, or explain how the techniques developed here could extend to other domains. Is there a real-world application where these methods could be used?

At a high level, the new DORA algorithm combines ideas from several existing approachs, in particular AlphaZero (Monte-Carlo Tree Search based methods), ReBel, and double oracle and regret matching methods. As the authors explain, this is a combination of various existing building blocks, but combined in a new way, where teh the MCTS based search is replaced with a regret matching approach, and where the policy network acts to propsoe actions for the search, but not regularizes the search after; the authors also nicely discuss the connection to Nash-Q learning. All in all, I think the algorithm is very interesting, and I wish the authors had investigated at least several small additional domains to see whether they get a good advantage there (again, to me Diplomacy is a very interesting domain, but to show true value you need to show the approach generalizes further, even to multiple simple games).

Empirically, I also find the results a bit mixed (or rather, the current write-up overclaims with respect to what the results actually show). The two player case is a two player zero sum game. For such games, computing a Nash equilibrium is polynomially solvable (using for example a very simple linear program). In Diplomacy, the action space is enourmous, so this direct approach does not work, but there are other tractable alternatives. Hence, I find this part of the paper a bit less interesting or surprising. The 7 player variant does indeed look far more challenging, but in this case, the results are mixed. The more realistic scenario is the one where you can place one of your agents in a tournament, but the remaining players would be controlled by other players (it is unlikely that you would get to control six out of the seven players in each game). And in this scenario the performance of the agents is not as strong. The authors consider this some form of an equilibrium selection problem, but I'm not sure this is really the case. Either way, it'd be good to include multiple existing bots in such tournaments.

Overall I quite like the paper, as it proposes interesting algorithms. But the claims seem a bit to strong to be justified by the existing results. Also, the key success here is not constructing much stronger agents that those available before, but rather doing so without imitation learning, in a nice algorithm that draws inspiration from existing approaches, and I'd feel much more confortable with this kind of a presentation.

The paper is reasonably readable (in particular the discussion of existing methods). It would be good to have a high level figure of the building blocks of the algorithm, and a table comparing this DORA agent with existing approach (the key building blocks they use, whether it is a supervised learning method or RL or a combination, does it use human data, runtime etc).



**Time Spent Reviewing:**

6 hours

---

> ### Author Response · Authors · 2021-08-09
> **Author Response**
>
> >**...I think the authors should extend the discussion of why this domain is so important, or explain how the techniques developed here could extend to other domains. Is there a real-world application where these methods could be used?**
>
> We will add to the introduction a motivation for our research and potential future applications, though to be clear we view our research as fundamental without a specific application in mind. We believe this research will eventually be applicable to all multi-agent domains involving both cooperation and competition. Potential future applications include machine negotiation and traffic navigation, to name just two.
>
> >**The 7 player variant does indeed look far more challenging, but in this case, the results are mixed. The more realistic scenario is the one where you can place one of your agents in a tournament, but the remaining players would be controlled by other players... And in this scenario the performance of the agents is not as strong. The authors consider this some form of an equilibrium selection problem, but I'm not sure this is really the case.**
>
> As Reviewer MTQy pointed out, we believe the fact that DORA achieves superhuman performance in two-player Diplomacy when training from scratch while only achieving moderate performance when training from scratch in 7-player Diplomacy is actually a very important scientific result that is important to share with the AI research community, rather than being a weakness of the paper. We agree that the setting of a single agent playing with six other agents (and ideally humans) is the most important setting; we use the six-DORA-vs-1-other-agent results primarily to provide additional evidence that equilibrium selection is a problem rather than as a measurement of DORA's strength. While our results do not conclusively point to equilibrium selection as being the culprit, it does provide very strong evidence in that direction (and we sought to phrase our conclusions in the paper in this way).
>
> >**Either way, it'd be good to include multiple existing bots in such tournaments.**
>
> All prior competitive Diplomacy bots have been trained on human data, and typically the *same* set of human data, so it is not clear to us if such a tournament would provide additional value and it may distract from the main results of the paper.
>
> >**...the claims seem a bit to strong to be justified by the existing results... ...the key success here is doing so without imitation learning... ...and I'd feel much more comfortable with this kind of a presentation.**
>
> Thank you for the feedback -- that is indeed what we intended to present and claim, and we will strive to revise the writing to reflect this.

---

### Official Review · Reviewer_xC8k · 2021-07-12

**Rating:** 6
**Confidence:** 3

**Summary:**

This paper describes an algorithm that trains agents through self-play with no human data and no reward shaping, and can accommodate the large action space of Diplomacy. It is an algorithm for action exploration and equilibrium approximation in games with combinatorial action spaces. This algorithm simultaneously performs value iteration reinforcement learning while learning a policy proposal network. Meanwhile, a double oracle step is used to explore additional actions to add to the policy proposals. They train an agent completely from scratch for a 2-player variant of Diplomacy and show that it achieves superhuman performance. They also train an agent for the 7-player no-press Diplomacy entirely from scratch with no human data for the first time and show that this agent differs radically from past agents that required human data. This approach therefore opens up the ability to investigate novel ways of playing the game.


**Limitations And Societal Impact:**

yes

**Main Review:**

The idea of approximating Nash equilibrium limited to a candidate actions set and leveraging a modified double oracle mechanism to introduce novel actions for exploration is original and feasible to deal with large action space. In experiments, their agents trained without human data achieve superhuman performance in a 2-player variant of Diplomacy and converge to an equilibrium in 7-player no-press Diplomacy. While some intuitive explanations are given, the theoretical support for the proposed approach is still weak. The following are comments that the authors need to address.

Major comments:

In line 170, as you regress the policy proposal network toward the NE policy profile using cross entropy loss, why do you need the policy proposal network? Why not directly use the NE policy profile to generate a candidate action set?

In line 160 a candidate action set at state s is generated by sampling N_b actions from policy proposal network \pi and selecting the N_c actions with highest likelihood. Why not just sample N_c actions from \pi?

As described in Section 4.2, the model closely follows the encoder-decoder architecture of [9]. Beside replacing graph convolution-based encoder with a simpler transformer encoder, the adjustments to the model architecture contain removing dropout layers and using seperate policy and value networks. Therefore, in Section 5.1, more ablation studies are required to distinguish the contributions of these adjustments to the performance of the agent.

In line 311, you mentioned that “in a competitive multi-agent game where agents must temporarily ally and coordinate, one expects that there may be a large number of different equilibria that are mutually incompatible”. Is there any theoretical support for this statement? It is better to provide more analysis and a more concrete image of the different equilibrium.

Since the experiment on the 2p0s variant of Diplomacy is one of the main results, more detailed information about the game (FvA) should be provided, such as the number of locations, the average number of possible actions per turn, etc.

Minor Comments:

Why is there a discount factor in the update rule in equation (5), but not in the definition of state value function in line 86?

The symbol v in equation (4) is undefined.

The equation (5) is confusing. According to line 154 where you redefine r to be a function of just the next state, r(s) should be r(s’).

The term “2p0s games” is used several times without definition. Even if I can get it could be 2-player zero-sum games.

In the second line of Algorithm 1, why does the function SAMPLEACTIONSFROMPROPOSALNET() include N_p as an input? As you described in line 191, N_p should be the number of actions in the pool generated in DO.

In line 242, more description for the first actor-critic method based exploiter agent is required.

In the column of Austria in Table 1, it is strange that the variance-reduced average score has higher standard error than the raw average score.

In Table 3, what kinds of Architecture is used? If Transformers are used, the result seems not better than those in Table 2.

In Table 4, it is unclear about the architecture of Model 1 (a trained value network with a fixed proposal network).

In line 486, “It is essentially identical to the architecture in [10]”, the main paper is [9] ( see line 226).

In Appendix B, the theoretical convergence of Deep Nash Learning in 2p0s games is only guaranteed in the tabular setting and when the candidate set contains all legal actions. However, under the condition of large action space, these assumptions appear to be contradictory to the main problem this paper likes to solve.

In line 511, based on the Bellman equation in reinforcement learning, the relation between Q and value function should be Q(s,a)=r(s,a)+V(f(s,a)) instead.

The statement in line 573 is unclear. What is the relation between “finding a best response for only one player at a time” and “recomputing the NE”? Which step in the algorithm does “recomputing the NE” refer to?

The paragraph from line 136-142 is the same as Section 2.2 in [9].


**Time Spent Reviewing:**

10 hours

---

> ### Author Response · Authors · 2021-08-09
> **Author Response**
>
> >**Why do you need the policy proposal network?**
>
> The policy proposal network allows us to sample a small set of likely candidate actions from Diplomacy's enormous action space and compute an equilibrium among this restricted set of actions. The NE policy profile is only feasible to compute in the first place because the policy proposal network has been used to restrict the action space.
> The policy proposal network is then regressed towards that NE profile, so that it becomes better in the future at proposing candidate actions that have support in the equilibrium. This is similar to how the policy network in AlphaZero massively prunes the action space to make search more effective, and then is trained to predict the results of that very same search, so that it becomes better at doing so in the future. We may have misinterpreted this question, so if it is still unclear please let us know.
>
> **Regarding how to sample actions (line 160):**
>
> Sampling $N_c$ actions directly from $\pi$ is a valid option, but it might result in sampling many low-probability actions that are not actually played in the equilibrium while missing actions that should be part of the equilibrium. Instead we use sampling to approximate the top $N_c$ actions from $\pi$, as the top $N_c$ are more likely to be part of the equilibrium than a random $N_c$.
> This is the same approach as in Gray et. al. [9]. Sampling is cheap but equilibrium computation is expensive, so it makes sense to have a large number of samples, then restrict that number to only the most likely for equilibrium computation.
>
> **Model architecture ablations:**
>
> We already provide comparisons for combined vs separate policy and value network in Table 2 (last two rows).
> Avoiding dropout is a common technique in RL, e.g., it was also used in Gray et. al. [9], so we are not sure this will add much to the paper, but we can run this experiment if needed.
>
> **Theoretical support for multiple equilibria in coordination games.**
>
> Yes, it is known that non-2p0s games admit multiple incompatible equilibria, and there is a large body of work studying incompatible equilibria in coordination games (Battle of the Sexes, Stag Hunt, and Majority/coalition games are relevant stylized examples) and some analysis of the behavior of learning algorithms in small matrix games with multiple equilibria [29].  However, there is less work showing the presence of multiple equilibria in large-scale games due to the difficulty of measuring this, and prior work has shown that self-play does not find human-incompatible equilibria in another large non-2p0s game, 6-player Poker [8]. We will add these references to clarify our statement, and add an example in the Appendix showing an example Diplomacy situation where a coordination subgame takes place.
>
> **Description of FvA Diplomacy:**
>
> We will expand the description of FvA Diplomacy to include more details. The entire map and therefore number of locations is identical to the 7-player game. The average number of possible player actions is similar to or larger than the number of actions in the full game because the players may control just as many units spread out across just as many locations, it's just that only 2 players control those units instead of 7.
>
> **Minor Comments:**
>
> Thank you for the numerous and thorough minor comments! Many of the things you mention are indeed variously typos, or inconsistent formulation, or omitted definitions, or points where more explanation could have improved clarity (e.g. discount factor, "2p0s" = 2-player zero-sum, $r(s')$, the instance of $N_p$ in Algorithm 1 you mentioned should be $N_c$, etc). We will correct these and the others you mentioned.
>
> To give additional clarification on a few minor comments:
>
> **Variance-reduced average score has higher standard error than the raw average score.**
>
> This is an artifact of the agent being lucky enough to win 100% of games as Austria, thus the empirical sample variance is technically 0 -- anomalously low -- so *any* nontrivial control-variate method will report a higher standard error. We are open to suggestions on a better presentation for these results.
>
> **Regarding Table 2, Table 3, Table 4, architecture and Elos**
>
> The Elos in these tables were computed on separate sets of games and are not intended to be comparable across tables, we will make this more clear. Architecturally, Model 1 is also GraphConv, but uses a weak imitation-learning policy and what are likely worse hyperparameters from early in development. It is intended just as a weak baseline agent.
>
> **Regarding Appendix B**
> Correct, this is just to establish the soundness of the underlying form of Nash-based value iteration, prior to the approximations needed for RL to be effective in practice (function-approximation via deep neural net to handle large state space, sampled double oracle and policy proposal net to find likely equilibrium actions to handle large action space).
>
> [8] Noam Brown and Tuomas Sandholm. Superhuman AI for multiplayer poker. Science, 2019.
>
> [9] Jonathan Gray, Adam Lerer, Anton Bakhtin, and Noam Brown. Human-level performance in no-press diplomacy via equilibrium search. ICLR 2021.
>
> [29] Michihiro Kandori, George J Mailath, and Rafael Rob. Learning, mutation, and long run equilibria in games. Econometrica: Journal of the Econometric Society, 1993.

---

### Official Review · Reviewer_k87e · 2021-07-13

**Rating:** 8
**Confidence:** 4

**Summary:**

Diplomacy is a complex 7-player strategy game that poses a challenge for RL due to its simultaneous moves and combinatorially large action space. In recent years significant progress has been made on learning human-level play by starting from imitation learning on a dataset of human play from websites, and then using some policy improvement algorithms. This paper tackles the exploration problem, introducing a policy improvement algorithm that reaches competitive-quality play without requiring a dataset, using Nash Q-learning, a double oracle-like method, regret matching, and geographically local perturbations for action proposals. They also find advantage from switching to a transformer-based architecture.

Their method (DORA) exhibits strong results in 1v1 human play, as well as being the first from-scratch agent able to play on par with DipNet (an imitation baseline) in a 1v6 setting, in a metagame that DORA wasn’t trained in.

Separately, their policy improvement method on top of an imitation network beats previous SOTA (SearchBot).

**Limitations And Societal Impact:**

The authors have adequately addressed the limitations and potential negative societal impact of their work.

**Main Review:**

This paper addressed a significant shortcoming in the existing literature: the equilibrium selection aspect of Diplomacy could not be explored because competitive algorithms couldn’t get off the ground without human play as a starting point. Therefore, any moves that were outside the human play distribution could not be explored effectively. This is an important contribution. It’s not very surprising, but it is great to see that a clever type of local perturbation is able to find good action proposals and explore the action space.

The paper established strong results in three regimes of no-press Diplomacy: France v Austria, 7-player Diplomacy within the human metagame, and 7-player Diplomacy “from scratch”. These need to be understood separately.

The 1v1 result is very convincing, being evaluated in a well-explained way using expert human players, and surpassing them overall. The paper tackles the key problem for this game variant.

The 7-player from-scratch result is also well-presented and impressive, and novel, building on a novel sampling-based generalization of double oracles to reach a new equilibrium in which existing human-data-based agents are beaten, while still performing reasonably well in the human-data metagame.

The seeming SOTA result using human data (rather than the double oracle action exploration method) is also very interesting, and the paper would benefit from more data and exposition of what makes up that substantial advantage over SearchBot; for example whether it’s an improved policy at inference time (with comparable computational cost), or whether it’s due to the Deep Nash Learning on top of the imitation network.


**Time Spent Reviewing:**

5

---

> ### Author Response · Authors · 2021-08-09
> **Author Response**
>
> >**...for example whether it’s an improved policy at inference time (with comparable computational cost), or whether it’s due to the Deep Nash Learning on top of the imitation network.**
>
> We believe that it is a combination of both. In early experiments we trained only the value network via self play, but used the human blueprint policy during both the training and the evaluation. We later added the policy loss to match the search policy and found that it improves the performance further. Note that in both cases the blueprint policy was used to generate candidate actions during training as mentioned in Appendix (line 632).
>
> Thanks for the feedback, we will add more discussion on 7p performance in the appendix.

---

### Official Review · Reviewer_4wnr · 2021-07-15

**Rating:** 6
**Confidence:** 3

**Summary:**

This paper builds off of Gray et. al. by swapping out the blueprint that is trained on human data with one that is trained from scratch using a deep learning variant of Nash-Q learning. The new method is able to achieve what appears to be superhuman performance in 1v1 Diplomacy and also does well but with mixed results in 6v1 Diplomacy against existing methods.

**Main Review:**

(This review has been edited. Initially I thought that the performance on 1v1 was not superhuman because I thought the authors just played against some top players online, but after looking through the supplemental materials they played against multiple players including the top rated player in the world and had a winning record against all players. I think this qualifies as being superhuman, and for that reason change my review to accept. I should have verified this earlier, my apologies.)

Diplomacy has emerged as a new benchmark in the field, and is a very large and complicated game. The authors are able to do well on this game, which is impressive. Most importantly, they are able to achieve superhuman performance on 1v1 Diplomacy for the first time.

However, as described in my summary of the paper, it seems that the main contribution of this paper is changing the blueprint strategy of Gray et. al. and replacing it with a deep learning version of Nash Q learning. The resulting method achieves about the same performance as the Gray et. al. paper. Although the Gray et. al. paper does not compare performance in 2-player games, my guess is that it would do similarly well as DORA did.

I don’t think the methods introduced are extremely novel or general. They seem to be modifications of existing methods (Nash Q Learning) to get it to work on Diplomacy. With that being said, the performance on two-player Diplomacy appears to be superhuman. Even if I suspect that SearchBot might also have been superhuman at 1v1 Diplomacy, they do not include these results in their paper.

I am still ambivalent about this paper. The method is not very novel and it is not clear that other methods would not have worked on this domain. As a result, I am not sure how others can build on this work. The authors do not present this method as a general method that will work well for any 2p0s Markov game, to do so they should have compared directly against other algorithms across a number of games.  However, this is the first paper to achieve superhuman performance on a large 2-player zero-sum game that has been popular with the research community, and for that reason I vote to accept.

____________________
I have read the rebuttal and the other reviews and do not change my score.


**Time Spent Reviewing:**

5

---

> ### Author Response · Authors · 2021-08-09
> **Author Response**
>
> **Re: superhuman performance in 1v1 Diplomacy**
>
> We thank the reviewer for adjusting their review due to the experimental results. We will try to make the 1v1 human experiments more clear in the main text.
>
> >**...it seems that the main contribution of this paper is changing the blueprint strategy of Gray et. al. and replacing it with a deep learning version of Nash Q learning.**
>
> A major contribution of this paper is eliminating the need to rely on human data in order to cope with the combinatorial action space of Diplomacy by instead introducing a novel action exploration algorithm. This is in contrast to [21,2,9], which all required human data to initialize a starting policy.
>
> >**The resulting method achieves about the same performance as the Gray et. al. paper.**
>
> This is incorrect. As shown in Table 6, our method outperforms Gray et al. [9] in 7-player Diplomacy when given access to the same human training data (HumanDNL). 1 HumanDNL agent playing against 6 SearchBots achieves an average score of 28% (a "tying" average score is 14.3%). In contrast, 1 SearchBot playing against 6 HumanDNLs scores 3%.
>
> However, perhaps more interesting is how our algorithm performs when trained *without* human data (DORA), which our method is the first to demonstrate competitively. In this case, we find evidence that it learns "strong" policies that are incompatible with a human population of agents: 1 SearchBot scores only 0.1% against 6 DORAs, while 1 DORA scores only 1.2% against 6 SearchBots. As Reviewer MTQy highlighted, this suggests that self play alone may lead to solutions that are incompatible with human play, which was not the case in prior multi-agent AI benchmarks like multiplayer poker. This validates Diplomacy as an important testbed for multi-agent AI research.
>
> >**Although the Gray et al. paper does not compare performance in 2-player games, my guess is that it would do similarly well as DORA did.**
>
> While we disagree with this guess, we wish to emphasize that the goal in the two-player experiments was to achieve superhuman performance from scratch without human data, and thereby demonstrate that our technique does not require human data in order to handle the combinatorial action space of Diplomacy.
>
> [2] Thomas Anthony, Tom Eccles, Andrea Tacchetti, János Kramár, Ian Gemp, Thomas C Hudson, Nicolas Porcel, Marc Lanctot, Julien Pérolat, Richard Everett, et al. Learning to play no-press diplomacy with best response policy iteration. NeurIPS 2020.
>
> [9] Jonathan Gray, Adam Lerer, Anton Bakhtin, and Noam Brown. Human-level performance in no-press diplomacy via equilibrium search. ICLR 2021.
>
> [21] Philip Paquette, Yuchen Lu, Seton Steven Bocco, Max Smith, O-G Satya, Jonathan K Kummerfeld, Joelle Pineau, Satinder Singh, and Aaron C Courville. No-press diplomacy: Modeling multi-agent gameplay. NeurIPS 2019.

---

### Official Review · Reviewer_MTQy · 2021-07-16

**Rating:** 7
**Confidence:** 5

**Summary:**

The paper proposes learning methods for the No-Press variant of the board game Diplomacy.

The method uses an equilibrium search method (similar to [9]) to solve for Nash in a single turn of the game, estimating future returns with a value function one turn in the future. Then a policy is trained to predict the equilibrium found, while the value function is trained to predict the values of the Nash equilibrium, which implements value iteration.

The equilibrium search method is based on prior work [9]. This tackles the large action space by restricting search to only the most likely actions from a policy network. This is extended in this work by using Double Oracle to expand the action set, and using a Diplomacy-specific method to generate proposal actions, which exploits the graph structure of the game. The paper also introduces a transformer-based model for learning to play Diplomacy, which outperforms previous Graph-Network approaches, and advances methods to estimate the exploitability of a Diplomacy agent.

Experimental results show that this results in an agent that can defeat human opponents convincingly in a 2-player variant of Diplomacy.


**Limitations And Societal Impact:**

Yes, except for points raised above

**Main Review:**

This paper is well written and easy to follow. The methods used are appropriate to the domain, and experimental data supports the claims of the paper well. I like that the human challengers were given an opportunity to practice against the agent before playing the test games.

It is a shame that no tester played many games - I assume this also means they played only some practice games too - as I’d be interested to see whether a player could improve against the bot with much more practice.

I didn’t find the techniques proposed particularly surprising, the paper seems to represent a judicious combination of existing ideas, rather than major algorithmic innovation.

However, Diplomacy is a very challenging domain, and there are many details that were considered here to perform well in it, which future researchers could build upon. In so doing, this paper provides useful empirical evidence about the efficacy of those methods in an important domain.

I particularly value that this paper demonstrates that the following are true:
1. Existing methods are sufficient to achieve superhuman results in the 2-player zero-sum version of Diplomacy, learning from scratch.
2. The same methods are (likely) unable to do the same for the otherwise similar 7-player variant. As the paper notes, this was not the case for many-player poker.

This motivates and enables researchers to extend beyond 2-player zero sum in an interesting domain. I think that moving beyond 2p0s would be of considerable value to the field.


Other points:

Are the results from single training runs of the methods, or from multiple seeds? If the former, can additional replications be included? Are the standard errors only over measurements, or also over repetitions of the training procedure?

I’m not a fan of the name “Deep Nash Learning”, because multiple algorithms already exist that combine deep learning with theoretically sound approaches to finding a Nash equilibrium. Perhaps something like Deep Nash Value Iteration would be more descriptive?

In Table 6, could you include your best human imitation learning agent? Several works since DipNet have reported improvements on DipNet, and it would be interesting to see how the state of the art for Imitation learning compares.

The conclusion that the method allows non-human equilibria to be explored in 7-player Diplomacy misses the more important point that incompatible equilbria exist and are a problem for learning based approaches, which hadn't previously been demonstrated. It is not clear that more than one alternative equilibrium can be found by DORA, though.

Line 492 (appendix) and later: citation maybe should be of [9], not [10]? The citation in line 653 has a similar off-by-one error.

The comment in lines 573-576 clarifies what I had thought was a typo in Algorithm 1. It might be worth including a comment in the main text? In the 7 player version, was this still done player-by-player, as the sampling for regret matching and potential BR value estimation means that this comment is perhaps no longer valid?

For the camera-ready version, could the HumanDNL experiment be repeated with the same settings as the from-scratch version?

Are there any insights to be had about the agents’ play styles, either in 1v1 or 7-player Diplomacy?

**Time Spent Reviewing:**

2.5

---

> ### Author Response · Authors · 2021-08-09
> **Author Response**
>
> >**Are the results from single training runs of the methods...**
>
> Thanks for the question - yes, the results are from single training runs. For camera-ready, we can certainly train one or two replications and will report their results as well. We do not expect the overall conclusions to be affected.
>
> >**...the name Deep Nash Learning**
>
> We can change "Deep Nash Learning" to "Deep Nash Value Iteration", thank you for the suggestion!
>
> >**In Table 6, could you include your best human imitation learning agent?**
>
> We will add our best human imitation learning agent to Table 6 and also add a version of SearchBot where the blueprint is set to that imitation learning policy.
>
> **Highlighting the point that incompatible equilibria exist and are a problem.**
>
> We agree that a major contribution of this paper is providing evidence that incompatible equilibria exist in Diplomacy and that therefore it may be a good domain for investigating techniques to address this problem. We will revise the writing in the paper to emphasize this point, and as part of training replications as discussed above, we will add an additional experiment directly measuring the incompatibility of the equilibria found between those independent replications.
>
> **Citation numbering error**
>
> Thank you for pointing out this mistake, we will fix it.
>
> >**In the 7 player version, was [DO] still done player-by-player?**
>
> Yes, DO is still performed player-by-player in 7p Diplomacy. Despite the sampled RM, it is still cheaper than computing BRs for the large number of candidate actions. We will clarify this in the text.
>
> >**For the camera-ready version, could the HumanDNL experiments be repeated with the same settings as the from-scratch version?**
>
> Yes, we will report results for HumanDNL with the same architecture as the from-scratch agent (DORA).
>
> >**Are there any insights to be had about the agents’ play styles, either in 1v1 or 7-player Diplomacy?**
>
> From speaking with top players who have played against our 1v1 bot, it appears that the biggest difference between DORA and top human players is that DORA is less risk-averse than human players, and that this works in the bot's favor on average. DORA also uses slightly different opening moves than humans typically do.
>
> The biggest differences between DORA and humans become apparent in 7-player Diplomacy. DORA matches recommended human opening moves for some powers, but for others might make radically different openings. For example, as Austria DORA may open by attacking Italy, which is widely considered a poor opening move in the human metagame. DORA also appears to be quite good at maintaining a balance of power within its equilibrium -- few self-play games end in outright victory within the first 35 in-game years.

---

### Decision · Program_Chairs · 2021-09-27

**Decision:**

Accept (Poster)

**Comment:**

The paper presents a self-play algorithm to learn to play Diplomacy entirely from scratch. Diplomacy is an extremely large challenge domain and as such, has previously required human data to perform competitively. The paper presents DORA which addresses these challenges with various solutions inspired by the literature: a basic change to Nash Q-learning to use (state-based) value functions, double oracle to address/prune the action space, and an equilibrium search. The paper lacks in novelty and breadth of applicability due to simply combining various known methods together for a focused application to a specific domain. Still, the domain in question is a particularly interesting one and has garnered recent interest from the MARL community. Also, the results themselves are novel, showing that DORA is enough to reach super-human in the two-player zero-sum variant without human data, but the same method applied to n-player had mixed success (unlike the case of poker). These results will be of significant interest to MARL researchers, and the paper makes valuable contribution to the community.